# Organ of Corti vibrations are dominated by longitudinal motion in vivo

Sebastiaan W. F. Meenderink [1] & Wei Dong [1,2✉]

Recent observations of sound-evoked vibrations of the cochlea's sensory organ of Corti (ooC) using optical coherence tomography (OCT) have revealed unanticipated and complex motions. Interpreting these results in terms of the micromechanical inner-ear processes that precede hair-cell transduction is not trivial since OCT only measures a projection of the true motion, which may include transverse and longitudinal displacements. We measure ooC motions at multiple OCT beam angles relative to the longitudinal axis of the basilar membrane (BM) by using the cochlea's natural curvature and find that the relative phase between outer hair cells (OHC) and BM varies with this angle. This includes a relatively abrupt phase reversal where OHC lead (lag) the BM by ~0.25 cycles for negative (positive) beam angles, respectively. We interpret these results as evidence for significant longitudinal motion within the ooC, which should be considered when interpreting (relative) ooC vibrations in terms of inner-ear sound processing.

[1] VA Loma Linda Healthcare System, Loma Linda, CA 92374, USA. [2] Department of Otolaryngology – Head and Neck Surgery, Loma Linda University Health, Loma Linda, CA 92350, USA. ✉email: Wei.Dong@va.gov

The mammalian ear processes sound over a wide range of frequencies (e.g., 20–20,000 Hz in humans) and intensities (20 µPa–20 Pa; 120 dB), while maintaining the ability to discriminate tonal frequencies and intensities that differ by only 0.2% and 1 dB, respectively[1]. This remarkable performance is achieved predominantly by delicate micromechanical processes in the peripheral inner ear, or cochlea[2]. The high sensitivity, wide dynamic range, and frequency selectivity of our hearing all diminish in damaged or postmortem ears[3], but the underlying cochlear mechanics of the living ear are not fully understood. The cochlea is an elongated and spiraled conical structure (Fig. 1a) that is part of the bony labyrinth, located in the skull's temporal bone. It is partitioned into two fluid-filled tubes by the stiff basilar membrane (BM) that supports the organ of Corti (ooC). Sound, entering near the cochlea's base via the stapes, evokes a traveling wave (TW) that propagates in the longitudinal direction towards the apex. Systematic changes in several anatomical and biophysical properties along the cochlear length affect the local amplitude and propagation speed of the TW in a frequency dependent manner[4]. This creates a place-based spectral decomposition of sound, or tonotopic organization, along the longitudinal axis of the cochlea in which high frequencies maximally excite the cochlear base and low frequencies the apex. During TW propagation, the active outer hair cell (OHC) responses play a pre-eminent role in maintaining normal hearing, but our understanding of how they accomplish this is incomplete. More detailed information of sound-evoked vibrations along the BM and from key structures within the ooC is needed to determine how the cochlear micromechanics contribute to our exquisite sense of hearing.

Unfortunately, the internal cochlear structures are small and poorly accessible. Moreover, their responses are on a (sub-)nanometer scale and vulnerable to physiological insult[3]. This makes measuring in vivo vibrations extremely difficult, and until recently they were only recorded from superficial structures (e.g., BM) at a few longitudinal locations. Spectral-domain optical coherence tomography (OCT) overcomes several of these limitations and has facilitated vibration measurements from structures within the ooC[5,6] that revealed several unanticipated and complex motions[7–12]. Although only separated from the BM by 20–100 µm, OHC-responses differed substantially and were more complex than the BM response: they had larger amplitudes and different phases, displayed wideband (hyper-)compression, and exhibited more rectification and distortion products. The interpretation of these observations is, however, frustrated by the measurement technique in that it only measures the component of the displacement that is projected onto the system's optical path. This methodological limitation is severe when interpreting vibrations within the ooC. While BM vibrations are largely restricted to the transverse (i.e., the cross-sectional) plane[4], ooC structures can also move in the longitudinal direction, in-and-out of this plane. Such motion is expected from hydrodynamical descriptions of the cochlea[13,14], but has only been observed in excised cochleae[15], with indirect evidence in recent in vivo experiments[7]. Without knowing the motion directions[16], any interpretation of the observed complex motion patterns becomes tentative. Measured amplitude differences may reflect differences in motion direction rather than in true amplitude along the OCT optical path, and the same holds for the phase. More concretely, data from two points (e.g., the two ends of an OHC) may be interpreted either in terms of length changes or as a rotation of the structure. Without knowing the vibration directions, it is impossible to decide between the two.

In this manuscript, the aim is to establish whether sound-evoked vibrations in the OHC region of the ooC are also in the longitudinal direction. Such motion may affect the interpretation of ooC vibrations in terms of cochlear micromechanical responses that determine our ear's remarkable sensitivity and frequency

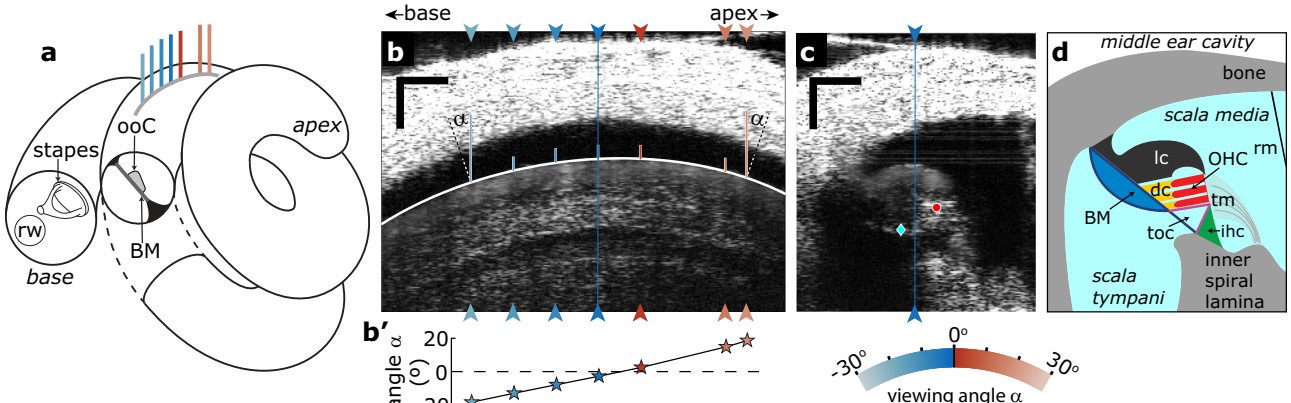

**Fig. 1 Different viewing angles in the 2ⁿᵈ turn of the cochlea. a** Schematic diagram of the spiral gerbil cochlea with a cross-sectional plane (orthogonal to the longitudinal axis) in the 2ⁿᵈ turn exposed to show the basilar membrane (BM) and the organ of Corti (ooC). The gray line represents the image location shown in **b** with colored lines indicating the different locations used for vibrometry. **b** Intensity image along the longitudinal axis in the 2ⁿᵈ turn of the gerbil cochlea. Arrowheads indicate longitudinal locations at which cross-sectional images and vibrometry were recorded. Scale bars, 0.1 mm. Solid white line: ellipse fitted to the boundary of the lateral compartment of the ooC. The angle between the normal of this ellipse and the vertical OCT beam at each longitudinal location at which a cross-sectional OCT image was acquired (arrow heads) served as an estimate for the "viewing angle α" between the OCT beam and the BM in the longitudinal direction. **b′** Viewing angles α of the OCT beam at the longitudinal locations indicated in **b**. These angles are color coded, see arched color legend below **c**. The same color coding is used throughout the manuscript. Scale bars, 0.1 mm. **c** Example cross-sectional intensity image, orthogonal to the longitudinal axis, obtained at the location indicated by the blue line in **b**. Sound-evoked vibrations were recorded from the BM (blue diamond) and within the OHC region (red circle). The blue line locates the intersection with the longitudinal plane. View into the paper is along the longitudinal axis, towards cochlear apex. Scale bars, 0.1 mm. **d** Schematic of the anatomical structures for the cross section shown in **c**. Fluid-filled spaces within the cochlea are light blue. BM basilar membrane, dc Deiter's cells, ihc inner hair cell region, lc lateral compartment, OHC outer hair cell region, ooC organ of Corti, toc tunnel of Corti, rm Reissner's membrane, rw round window, tm tectorial membrane. The blue-red colored arch encoded viewing angle, the same scale is used throughout the manuscript.

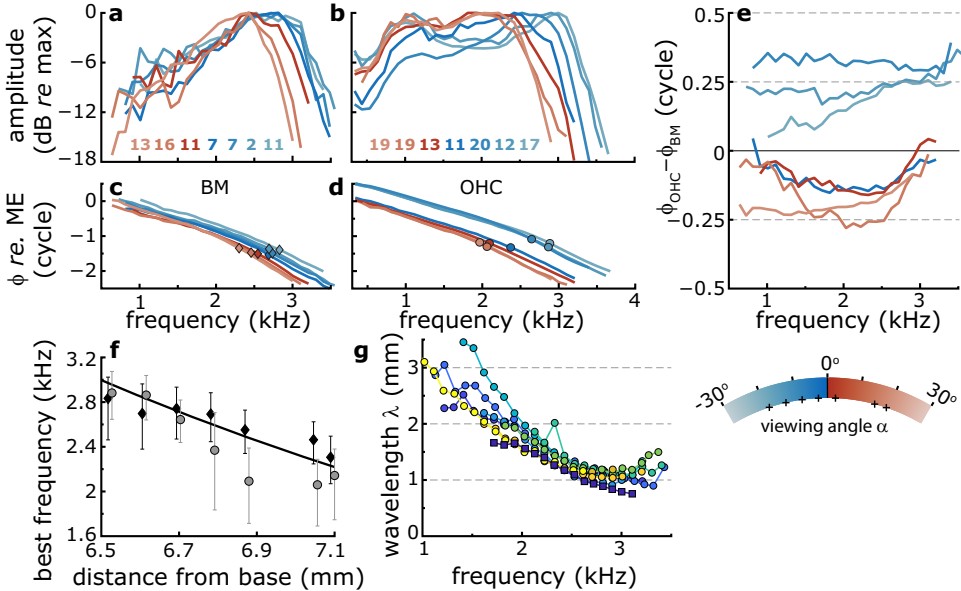

**Fig. 2 Vibratory responses along the longitudinal axis of the cochlea.** Relative amplitude of (**a**) BM and (**b**) OHC vibratory frequency response curves (FRCs) measured at different longitudinal locations in the gerbil cochlea. Different colors correspond to the color coding for recording location in Fig. 1b. Color-coded amplitudes (dB re. 1 nm) at BF are given in the keys. **c**, **d** Corresponding phase data, normalized to the middle-ear (ME) response. Diamonds and circles indicate best frequency (BF). **e** Phase difference between the responses in the OHC region and on the BM at each longitudinal recording location. **f** BF for the BM (black diamonds) and the OHC region (gray circles) determined from the FRC amplitude curves in **a** and **b**. Error bars give FRC-bandwidth at 1-dB below the BF response. The black line is a fit of the gerbil place-frequency map[18] to the BM-BFs. This fit was used to set the values along the abscissa. Stimulus level: 30 dB SPL per frequency component. **g** Wavelength of the TW. Here, each color represents a separate set of recordings that were obtained from $n = 6$ animals. We did not observe a dependence of wavelength on stimulus intensity. The square, darkest-blue symbols are from the phase data in **c**. The lines in **a–e** are color coded according to viewing angle $\alpha$ (see Fig. 1b') with the key below **e**. Small plus-symbols in the key indicate the viewing angles for these recordings.

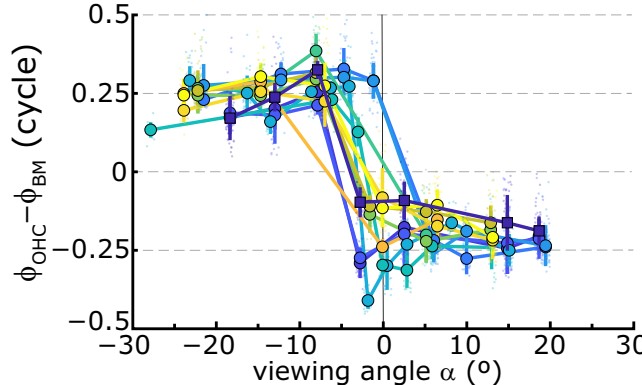

**Fig. 3 Viewing angle determines the phase difference between OHC and BM.** The different curves show $\phi_{OHC}-\phi_{BM}$, averaged across frequency, as a function of viewing angle $\alpha$ (see Fig. 1b') for different series of recordings. These were from $n = 6$ gerbils and were obtained at stimulus intensities between 30 and 70 dB SPL. Small colored circles give individual data points, error bars give ±1 s.d. around the mean. The square, darkest-blue symbols are for the recordings in Fig. 2.

selectivity. To determine if longitudinal motion occurs, we measured BM and OHC vibrations at multiple longitudinal locations in the middle turn of the gerbil cochlea to determine the phase difference, $\phi_{OHC}-\phi_{BM}$. This phase difference is predicted[7] to systematically and dramatically depend on the "viewing angle" between the OCT measuring beam and the BM in the longitudinal direction if the longitudinal (OHC) motion is out of phase with the transverse (BM) vibrations of the TW.

One method to vary the viewing angle is to rotate the OCT's beam direction relative to the inner ear[12], but anatomical

restrictions due to the cochlea's location make this difficult. The curved nature of the cochlear partition, however, offers a unique alternative to systematically vary the viewing angle in small steps without having to move the preparation or rotate the OCT (Fig. 1). By recording BM and OHC vibrations at multiple longitudinal locations, we exploit this curvature to determine how $\phi_{OHC}-\phi_{BM}$ depends on viewing angle, and whether it changes sign when the OCT beam is orthogonal to the BM.

## Results

With our OCT system we visualized a 0.6–0.8 mm stretch along the longitudinal axis of the second-turn gerbil cochlea (Fig. 1a, b). Across this section the cochlea coiled appreciably, resulting in considerable and systematic variation in the OCT viewing angle relative to the longitudinal axis of the BM (see Fig. 1b'; the viewing angle $\alpha$ varied between −28° and +19° across animals). At multiple locations along this axis, orthogonal (cross-sectional) images were obtained (e.g., Fig. 1c, d) in which sound-evoked vibrations of the BM and the OHC were recorded from two representative locations[8,11,12,17]. In each cross-sectional image, these two locations were chosen such that the distance between them did not systematically vary with viewing angle [$F_{1,43} = 2.73$, $p = 0.11$; distance mean (±s.d.) = 94 (±21) μm].

BM frequency response curves (FRC) were tuned (Fig. 2a) to best frequencies (BFs) that systematically decreased with distance from the base (Fig. 2f; diamonds). Using a cochlear place-frequency map obtained from gerbil auditory nerve fibers[18], these BM-BFs indicated that the recording locations were between 6.5 and 7.3 mm from the base of the cochlea. For each BM-FRC, phase (Fig. 2c) accumulated with frequency in a manner characteristic for a longitudinally propagating TW that slowed down when

approaching the location's BF. We calculated wavelengths (λ) from BM phase responses across the longitudinal recording locations (Fig. 2g) and found them to vary with frequency between λ ≈ 1 mm near BF and λ ≈ 2–3.5 mm below BF. These values are similar to wavelength estimates in the ~3-kHz region from cochlear-mechanical[3,19,20] and auditory-nerve data[21]. Like the BM responses, OHC-FRCs were tuned (Fig. 2b) with similar BFs that decreased with longitudinal location (Fig. 2f; circles), although they exhibited a substantially larger sub-BF response[7,11]. The accompanying phase curves (Fig. 2d) also showed location-dependent TW characteristics, but they differed from the BM phase data in that these curves were arranged in two groups that differed by ~0.5-cycle in their vertical offset. As a result, OHC responses either lagged or lead the BM by ~0.25 cycle (Fig. 2e). This results in a sudden change in the sign of $\phi_{OHC}-\phi_{BM}$ when plotted as a function of viewing angle (Fig. 3). This phase difference exhibited a robust trend across animals: for negative viewing angles, corresponding to the higher-BF longitudinal locations, the OHC responses led the BM by ~0.25 cycles, which abruptly transitioned to a quarter-cycle phase lag when the viewing angles were positive. We obtained recordings using stimulus intensities between 30 and 70 dB SPL but did not observe a systematic level-dependent effect [Fig. 4 and ref. [22]].

Since viewing angle and cochlear longitudinal location co-varied in these experiments, either can potentially underlie the observed effect. To discount the latter, we recorded $\phi_{OHC}-\phi_{BM}$ over a range of viewing angles twice in the same animal. After the first series of recordings (Fig. 4, black lines & circles), the animal was rotated 10° around the cochlea's modiolar axis such that a slightly more basal region of the cochlea was probed in the second series of recordings (Fig. 4, red lines & diamonds). Despite this 10°-rotation, the dependence of $\phi_{OHC}-\phi_{BM}$ on viewing angle was unaltered, discounting the possibility that it is the absolute longitudinal location that determines the observed phase phenomenon. In the same animal, we also obtained post-mortem recordings (Fig. 4, gray line & squares) at 80 dB SPL in which the OHC and BM moved almost in-phase independent of viewing angle. In two other animals (n = 3) in which postmortem data were recorded, we also did not observe the 0.25 to –0.25 phase transition that was present in vivo (n = 6) at stimulus intensities between 30 and 70 dB SPL.

## Discussion

The data presented here reveal a phase difference between the BM and OHC vibratory responses that strongly depended on the angle between the BM along the longitudinal axis of the cochlea and the incident OCT light beam (e.g., Fig. 3). We showed that the observed dependence is not from changes in the longitudinal recording locations (Fig. 4). Moreover, it is unlikely that local changes in the anatomy or physiology along the longitudinal axis would cause the effect. The 0.5-cycle phase transition is complete between points separated by only ~100 µm along the cochlear length, a spatial range in which changes in BM vibration are only small and gradual (e.g., Fig. 2a, c). In addition, the large change in of $\phi_{OHC}-\phi_{BM}$ would require a sudden change in the local anatomical and/or electro-mechanical ooC properties, which is not observed[23–26]. Finally, the same effect was observed in all animals (Figs. 3 and 4), despite differences in anatomy and experimental conditions/longitudinal location.

We interpret this result to indicate that sound-evoked OHC motions are primarily along the longitudinal axis, which effectively masks contributions from any OHC motion in the transverse plane. Descriptions of the hydrodynamics of the cochlea[13,14,27] predict that particle motion some distance away from the BM is along an ellipse, with displacements in the longitudinal direction that lag the transverse vibrations by 0.25 cycle. OCT measures the actual motion that is projected onto (parallel with) its optical beam. When the actual motion contains both longitudinal and transverse components, their relative contribution to the measured response depends on their relative magnitudes and the OCT viewing angle[7]. A simple geometric relation describes how $\phi_{OHC}-\phi_{BM}$ depends on both the true motion's "ellipticity" (i.e., how flat or round the ellipses are, given by the amplitude ratio of longitudinal and transverse motion) and the viewing angle of the OCT beam (Fig. 5b). Most notable in this relation is a 180° reversal of $\phi_{OHC}-\phi_{BM}$ when the viewing angle changes sign. We exploited the cochlea's natural curvature to vary this viewing angle (Fig. 5a). Accepting a few degrees of uncertainty in how well the calculated viewing angles describe the angle between the OCT beam and BM in the longitudinal direction, the data (Figs. 3 and 4) are qualitatively and quantitatively well described by the situation in which the projection of the longitudinal motion dominates the recorded OHC vibrations. Anisotropies in the ooC are likely to complicate the relatively simple hydrodynamical description of the cochlea and its predictions of (the orientation and shape of) the elliptical fluid motion. For example, the geometry of the OHC/Deiter cell and its phalangeal process/tunnel of Corti may favor longitudinal re. transverse vibrations, resulting in flatter ellipses than expected from the ratio between wavelength and wave medium depth alone. The rapid transition in the observed phase differences (Fig. 3) suggests that the ellipses within the OHC region are relatively flat, with much larger motion in the longitudinal than in the transverse direction. With this, it seems that the recorded OCT responses within the OHC region are largely dominated by longitudinal motion that mask any contributions from motion within the transverse plane to the measured response.

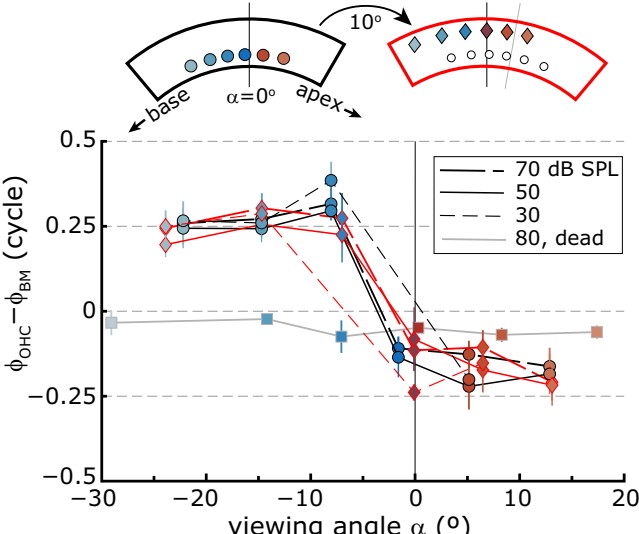

**Fig. 4 Viewing angle, not longitudinal location, determines $\phi_{OHC}-\phi_{BM}$.** OHC–BM phase difference, averaged across frequency, as a function of viewing angle α relative to the longitudinal axis of the BM (see Fig. 1b) for two series of recordings from the same animal, obtained at three different stimulus intensities (see legend). Vertical error bars give ±1 s.d. around the mean. After the acquisition of the first series (black lines, circles), the animal was rotated 10° around the modiolar axis such that a more basal region of the cochlea was visualized and a second series of recordings was obtained (red lines, diamonds). The cartoon at the top represents this rotation, with the viewing angles at the recording locations color coded according to the key in Fig. 1. Square symbols/gray lines are for data acquired postmortem in the same animal.

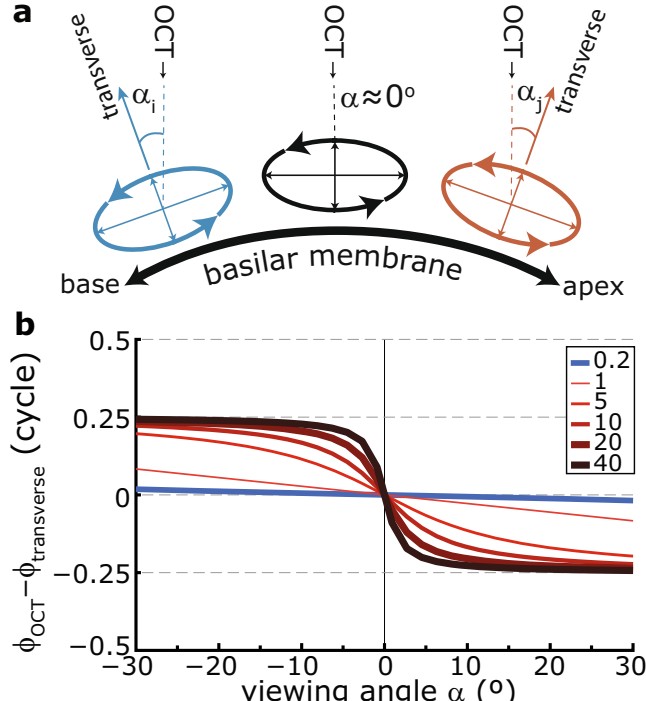

**Fig. 5 Effect of viewing angle on the relative phase $\phi_{OCT} - \phi_{transverse}$.**
**a** Due to the natural curvature of the cochlea/BM in the longitudinal direction (thick black line), a vertical OCT measurement beam (dashed lines) will view the elliptical motion in the OHC region at an angle ($\alpha$) that varies with longitudinal location within the cochlea (different colors). **b** Expected phase difference $\phi_{OCT} - \phi_{transverse}$ for elliptical motion (see text) with different aspect ratio, here, amplitude ratio of longitudinal and the orthogonal vibrations (see legend). For ratios >1 (i.e., a larger longitudinal component, red lines), the phase difference systematically varies between +0.25 and -0.25 cycles. The 0.5-cycle transition occurs more abruptly for larger amplitude ratios. When the orthogonal motion is larger (ratio <1, blue line), only a small phase difference occurs for the depicted viewing angles. Irrespective of amplitude ratio, the phase difference flips sign when $\alpha = 0°$.

Although this makes any interpretation of the complex motions observed within the ooC in terms of (relative) transverse or radial vibrations speculative, it may explain why, from all the different intracochlear vibrations that have been documented, BM responses are most like the tuned auditory nerve fiber (ANF) response[22,28,29]. It would not be surprising that when the longitudinal responses are quantified and accounted for, the apparent complexity of intra-ooC vibrations is much reduced such that they become more like the well-known BM and ANF responses.

## Methods

**Animal preparation.** The care and use of animals were in accordance with guidelines of, and approved by, the Institutional Animal Care and Use Committee (IACUC) of the VA Loma Linda Healthcare System. Cochlear vibrations were measured from the left ear in adult, female Mongolian gerbils (*M. unguiculatus*, $n = 6$) that were part of a larger study group[22]. Animals were anesthetized using intraperitoneal injections of a ketamine/xylazine (80 and 10 mg/kg, respectively) cocktail. Supplemental doses were administered to maintain areflexia. Core temperature was kept at ~38 °C using a heating pad (Harvard Apparatus). Animals were tracheotomized, but not actively ventilated. A ventrolateral surgical approach exposed the left bulla, which was opened to visualize the cochlea. The pinna and the cartilaginous ear canal were resected, and a probe containing a microphone and transducer assembly (ER-10X, Etymotic Research) was placed within a few mm of the tympanic membrane. Animals were not allowed to recover from anesthesia and were euthanized by anesthetic overdose at the end of the experiment.

**Optical coherence tomography and vibrometry.** A spectral domain OCT system (Thorlabs Telesto III TEL321C1 equipped with an LSM04 objective) that operated with a central wavelength of 1310 nm (bandwidth: 170 nm) was used to non-invasively image through the cochlear bone and record the vibratory responses in the middle turn of the exposed, but otherwise structurally uncompromised, cochlea. Acquisition of optical spectra was controlled by externally generated TTL pulses (rate: 27.9 pulses/ms) that were synchronized to the stimulus generation and microphone acquisition system (RX6: Tucker Davies Technologies system III). These spectra were converted into depth-resolved, axial information (A-line) using Fourier analysis.

Intensity images (B-scans) were constructed from 1024 A-lines obtained at fixed distances along a "scan line". To improve the quality of these images we averaged at least 60 B-scans, where each B-scan was completed prior to acquisition of the next one. The scan line followed by the OCT beam was either parallel (Fig. 1b) or orthogonal (Fig. 1c) to the longitudinal axis of the cochlea, where the latter yielded a cross-sectional image of the cochlear duct. The length of the scan line was 1.6–3 mm for parallel images and 0.6–1 mm for orthogonal images. For vibration measurements (M-scans), the axial phase information of the A-lines was used (phase-sensitive OCT, e.g., refs. [6,30]), assuming a refractive index of 1.3 for the intracochlear structures. Here, the OCT measurement beam was kept at a fixed position, and a series of time-stamped A-lines was recorded while an acoustic stimulus was presented to the ear. For M-scans obtained at different longitudinal locations, the angle between the vertical OCT beam and the ooC varied due to the natural curvature of the cochlea (Figs. 1b and 5a), which allowed us to assess the relative size of vibrations in the longitudinal and cross-sectional plane. Vibratory responses of the umbo were recorded immediately following each animals' death and served as a reference for the phase of the intracochlear responses to remove any delay within the experimental setup.

**Acoustic stimulation.** Acoustic stimuli consisted of 42 equal-intensity frequency components, each one with a random starting phase. Sound pressure levels were calibrated in situ and are expressed in decibels re. 20 µPa (i.e., dB SPL). Stimulus frequencies were irregularly spaced ("zwuis": see ref. [31]) and were chosen such that they all had a whole number of cycles over a "periodic block" of $w = 334,821$ samples (~12 s). Typically, 3–5 concatenated periodic stimulus blocks were presented (i.e., stimulus duration: 36–60 s). The stimulus was preceded and followed by a ~100-ms signal to accommodate ramps (5 ms, raised cosine) that turned on and off the tone complex, respectively. For a selected pixel (depth) along the A-line, the average response waveform (i.e., the pixel's OCT phase response versus time) across the periodic blocks was calculated (ignoring the 100-ms onset and offset) that was used for subsequent analysis.

**Statistics and reproducibility.** Vibratory responses were recorded from the left ear in $n = 6$ gerbils. Magnitude and phase of the vibratory responses were extracted using Fourier analysis. No artifact rejection was employed. Response amplitude and phase was plotted as a function of stimulus frequency (Fig. 2a–d). The response to a frequency component was considered above noise when a Rayleigh's test for uniformity indicated significant ($p \leq 0.001$) phase locking to the stimulus[32]. Best frequency (BF; Fig. 2f) was calculated for each frequency response curve (FRC) by fitting significant (above-noise) datapoints with frequencies >1 kHz with an 8th-order polynomial. Only when $r^2 \geq 0.9$, this polynomial was used to calculate BF as that frequency for which the largest local maxima between 1 and 5 kHz occurred. For each animal, the frequency dependent wavelength ($\lambda$; Fig. 2g) was calculated by fitting the phase FRC from multiple ($n > 5$) longitudinal recording locations with a line using least square minimization. The slope of this line is the reciprocal of wavelength; it is only reported when $r^2 \geq 0.8$. The angle between the OCT beam and the BM in the longitudinal direction (termed "viewing angle $\alpha$" in the manuscript) was taken as the angle between the normal of an ellipse fitted to the boundary of the lateral compartment of the ooC (Fig. 1b, solid white line) and the vertical OCT beam. The phase difference $\phi_{OHC} - \phi_{BM}$ (Figs. 3 and 4) is reported as the mean ±1 s.d. over all stimulus frequencies.

Custom software for stimulus generation, hardware synchronization, and all signal analysis (including statistics and linear regression) was implemented in Matlab (R2018b); control of the OCT system was programmed in C♯ (Visual Studio Dev14).

## Data availability

All data presented in this study are available for download at figshare (https://doi.org/10.6084/m9.figshare.18217127)(ref. 33).

## Code availability

All code for data acquisition and analysis was based on standard functions available in Matlab (R2018b) and C♯ (Visual Studio Dev14), and is available upon reasonable request.

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

## Acknowledgements
We thank Drs. Peter Narins, Hyle Park and Xiaohui Lin. This work was supported by NIH/NIDCD R01DC011506 and R21DC019998 to W.D.

## Author contributions
Both authors devised and performed the experiments and wrote the paper. S.M. developed software for data acquisition and analysis. W.D. obtained funding and managed the lab.

## Competing interests
The authors declare no competing interests.
