## [Peer Review File · Communications Biology]

Reviewers' comments:

Reviewer #1 (Remarks to the Author):

In this paper the authors described the use of OCT for measure the direction of vibration of basilar membrane (BM) and organ of Corti (ooC). It is a relevant topic to better understand the cochlear micromechanics contribution to hearing.

Despite the relevance of the subject the aim of this paper was not well defined/stated by the author, making it difficult to the reader to identify and follow the main results. By careful reading the paper this reviewer understands that the author goal was to demonstrate that the curvature of the cochlea could be used as a method to measure the vibration direction of motion. It would be equivalent to change the OCT angle-of-view from previous published works.

Another important aspect is related to the Methods described in the work. Due to the different steps and complexity of the analysis more details should be provided.

From this reviewer point-of-view this work must have major revisions before being considered for publication.

Further specific points are described bellow.

Abstract

1 – The authors stated: "...extended region in the low-frequency apex of gerbil cochleae...", the work described OCT images acquisition at the second-turn gerbil cochlea, not at the apex. Please clarify.

Introduction

2 – Please define BM and ooC on its the first use at the main body of the text.

3 – Clearly define the goal of the work at the end of the introduction section.

4 – The following fragment of the last paragraph of the introduction section of the manuscript "At near-perpendicular... ..observed motions within the ooC." describes findings from this work and should be placed at a different section (results, discussions e.g).

Results

5 – "BM frequency response curves (FTC) were tuned (Fig. 2A) to best frequencies (BFs)"

How do you define the BF? Should be clearly defined at the Methods section.

6 – "We calculated wavelengths (l) from BM phase responses across the tonotopic recording locations"

How was the wavelength calculated? Should be clearly defined at the Methods section.

Methods:

7 – "The scan path was traversed multiple (>10) times and the intensity profiles averaged"

Does it mean that multiple B-scans were acquired at the same position? How many exactly? If changed, what was the criteria for choosing? Please rephrase and provide more details.

8 – "Between M-scans, the angle between the..." Does it mean between adjacent M-scans?

9 – It is not clear how amplitude and phase was measured from the M-scan. Was it measured by the structural displacement? Please provide more details.

10 – It is not clear how BF was measured. Please provide more details.

11 – It is not clear how the wavelength was measured. Please provide more details.

Discussions

12 – The authors presented a hydrodynamic analysis of (Fig 4) to discuss the results. It is nor clear how de authors reached the conclusion using this model. One suggestion is to clearly define this analysis at the Methods and present its Results.

Reviewer #2 (Remarks to the Author):

The authors have used the natural curvature of the cochlear partition as a means to investigate the variation of the basilar membrane and OHC-region vibration as a function of the beam axis of an OCT-vibrometry measurement system. This is a clever approach because neither the preparation nor the OCT system need be physically moved to acquire these data. The authors provide arguments for why they are in fact measuring an effect that describes the vector nature of the velocity, and not some other confounding effect. These arguments are strong, and would be more convincing once data on the dependence of the angle of the optical axis with the BM is quantified when changing the longitudinal location (see below). The authors show that a frequency-averaged-phase shift consistent with an elliptical vibration path (i.e., with both in-plane and longitudinal vector components) is present – not just in-plane motion) is present – and these results are very interesting and potentially paradigm shifting.

The segregation of the positive and negative phase dependence with viewing angle is striking (shown in Figure 3A and in Figure 2E) and the authors have shown in great detail this phase result. In addition to these phase data, the amplitude of the motion as a function of the optical axis angle should be presented and analyzed with respect to consistency of the finding that there is significant longitudinal motion. This quantification may be accomplished by presenting the normalizing value of the amplitude in Fig. 2A and 2B for each angle/longitudinal location in tabular or graphical form. This would be followed by a discussion of whether these results support your conclusions. By presenting these normalizing data, it also serves to provide the amplitude of motion for archival purposes.

Change of optical axis with the BM/radial direction: The results could also change if the optical angle axis changes with respect to the BM (as shown for one optical axis by the green line and white dashed line for the radial direction in Fig. 1C). Providing an estimate for the change in this angle as a function of longitudinal location (likely a function of the helical coil angle of the cochlear ducts).

Describe the acceptance criteria for presenting the data (e.g., discuss rationale for choosing animal #1 for 30, 50, and 70 dB while animal #3 only for 50 dB).

Minor: Using FRC instead of FTC for frequency response curves makes more sense.

Overall, this is a clearly written article with a key result that could be very important to our field. I appreciate the clear analytic presentation of the results (no over-exaggeration of the results).

Reviewer #3 (Remarks to the Author):

Review of: Organ of Corti vibrations are dominated by longitudinal motion in vivo

The novelty of this study is that longitudinal motion of the organ of Corti at the apical region of the gerbil was measured with OCT by measuring transverse-radial motion of the BM and OHCs at different longitudinal locations. As viewing angle is changed (α in Fig. 1B), the phase relationship between OHC and BM changes due to the curvature of the cochlea. The measurements of motion from OCT showed that the motions of cochlear structures are dominated by longitudinal-directed motion from the traveling wave. This is analogous to the flat elliptical motion (lots more motion longitudinally than transversely as shown in Fig. 4 C), due to the long wave with respect to depth of fluid. This study is showing that motion of the cochlear structures measured by OCT is actually dominated by this longitudinally directed motion.

This experiment is elegant, the description is clear, and the results are of interest to hearing researchers who are concerned with the mechanical motion of the structures that affect transduction of sound.

Specific comments:

Figure 1: Recommend showing naked OCT without markers and lines in addition to what is shown. The lines and markers obstruct the view obtained by the OCT. Another picture just like C to the left of C without lines and labels. There's enough space in the lower row for this. The description in caption in (C) for the green line (A-line) is confusing, please clarify.

How do you know the relative angle α ? Also, these angle differences are small, how can you be sure of their accuracy?

Pg4, last para: Please be more specific in what you mean by "this": "At multiple locations along this axis,...". Also unclear: "...from two representative locations (8, 11, 12, 14)." What do the numbers 8, 11, 12, 14 refer to? Also what does F (and subscript numbers) mean?

Number of results are minimal: #1 (responses to 3 input levels), #2 (2 input levels), #3 (only 1 input level). All 3 animals are from different distances from base. Explanation would be useful.

Fig. 2: The legend does not have green, but the plots do. For quick interpretation, add labels "BM" to (A) and "OHC" to (B).

Fig. 2: responses are referenced to ME response. Would be good to show ME response re input sound pressure.

Fig. 2: Can "tonotopic axis" be better defined? Do you mean longitudinal axis?

Pg 5-6: Define sub-BF response. The second lower-frequency hump often seems just as big as the hump for CFs (2B). Is max magnitude always between 2-3kHz? The low-frequency hump at ~1kHz also looks just as big for most.

Pg. 6: Refer back to Fig. 1B to remind readers of what "viewing angle" α means.

Discussion:

Please define "tonotopic axis" (same for Fig. 2 caption). "Longitudinal" might communicate better.

Fig 4: Concept in A, B, C are classic motion of particles in water near the interface of air with a wave. It would be helpful to describe it that way. Also, when describing "depth", it's very confusing for that to be above. For A, B, C, it would be better to have it upside down. Also, it would be good to clarify in the caption that your interpretation of what is going on in the cochlea is like C.

How does the motion observed fit with other studies, such as those that focus on the anatomy of the cochlear partition, such as the phalangeal process of linking OHCs to dieter cells longitudinally and radially in a diagonal manner?

Reviewer #4 (Remarks to the Author):

The authors exploit the curvature of the organ of Corti to systematically vary the relative angle between portions of the OHC and BM regions, by translating their OCT beam along the tonotopic axis of the middle turn of the cochlea. In doing so, they noted a phase grouping in the OHC region measurements, which relative to the BM varied by a lead or lag of .25 cycles and depended upon the

angle of the measurement beam relative to the structures. The authors then employ a longwave model to demonstrate that such a phase transition would occur in the presence of longitudinal motion of the OHC region relative to the BM. They discuss and discount the possibility that these data may be derived from variations in the anatomy due to their use of multiple longitudinal locations as a means of varying the beam angle – which simplifies the experimental approach. They conclude by suggesting that longitudinal motion is a substantive component of organ of Corti mechanical function, and that the vector sum of OCT measurements recorded from the OHC region may be complicated by this additional, unconsidered component.

The experimental design is imaginative and the use of simple translation of the beam position may reduce the experimental error derived from manipulation of the preparation or beam angle. The ideas raised in this paper are potentially significant, thoughtful and intriguing, but more explanation and data are needed.

Major concerns and comments:

1. Providing an n of 3 for such a novel and significant finding is insufficient. Furthermore, providing a selection of sound pressure levels from each preparation with no justification for the selection criteria makes it more difficult to understand the full picture of the data. Within animal frequency data are analyzed statistically, but between animal data are not, because there is not enough.
2. The term OHC region is rather indefinite given that published studies are measuring vibrations of the reticular lamina (RL) and the base of OHCs at the Deiters' cell junction. These have different transverse and radial motion vectors. More specifically, from where are the measurements of this study?
3. The hypothesized elliptical motion is set against the fixed boundary of the RL and tectorial membrane (see 11 below for more about the boundary). Does this imply Rayleigh type water waves? In that case wouldn't the ellipse be oriented with the long axis in the transverse direction the greatest movement be near the basilar membrane?
4. Further to 3, would the nature and location of the elliptical motion be a way to test the hypothesis? Firstly, there would be a graded effect with depth. Secondly some Rayleigh waves have reversal of the elliptical direction with depth. See also point 11.
5. Figure 1 shows the direction of light beam view to be passing through the Hensen's cells when observing the OHC region. Hensen's cells are often the exact location of vibration measurements in studies because of the high degree of reflectance. Even when the OCT coherence gate is located at the OHC region the measured vibration can be a combination of the gate and of out of gate vibration. Hensen's cells could even dominate. Might Hensen's cells have a longitudinal vector component and view direction result in a phase flip? Resolving such questions requires more data on the transverse and radial motion of the OHC region.
6. The OHC region is said to be measured here and this phase transition occurs at the OHC place, one may surmise that longitudinal motion may be enhanced by amplification and as such the phase transition is a physiologically vulnerable process. No such data (such as post mortem measurements) was provided to confirm or refute this.
7. Further to 6, the phase transition shown for animal #1 in figure 3 indicates that the magnitude of the phase transition might show some level dependence. Since there is just one 30 dB SPL record shown, it is difficult to know if this is repeatable.
8. The authors declare that the phase flip is entirely dependent upon the sign of the viewing angle. This is not borne out by the data – animal #1 in figure 3a flips for a still negative viewing angle. It is difficult to tell if animal #2 flips with sign of the angle as there appears to be missing data around the 0 point for this animal. Animal #3 does follow the expected behavior. This variability highlights a requirement for more data presented more consistently across angle, level and frequency.
9. In figure 2 E, the relative phase between the two structures is shown as a function of frequency. For each curve shown there is a phase variation as frequency increases, and the direction of this variation depends upon the sign of the viewing angle. For negative angle measurements, the high frequency phase difference trends towards 0, so the two structures now move in phase. However for positive angle measurements the same phase difference is now magnified. If there is a longitudinal component, there is clearly some frequency based variation of its behavior. Is it possible to reconcile

the angle sign dependent widening phase difference vs converging phase difference observed here?

10. Also, looking at Fig. 2F, one can see that the BF for the BM and OHCs differs in a rather dramatic fashion. It seems to be lower for the OHCs. The problem here is that the legend says the data were acquired at 30 dB SPL, but only one animal had data at this sound pressure level. So despite the error bars, it seems to be an n of 1 here? And then the linear fit is calculated for all the points. In short, this is rather confusing and adds to the need for more experiments.

11. The authors state that elliptical motion where the major component is longitudinal produces a shallow wave profile, limiting the extent of the wave's pressure influence in the transverse direction to the boundaries of the organ of Corti. Is the effect subject to a form of scaling symmetry? Would the shorter wavelength of a higher frequency wave still be dominant to the transverse displacement? If so, the transverse wave depth would be even smaller. How can this be explained in the context of the observed motion of Reissner's membrane, which is assumed to follow the motion of the organ of Corti? Additionally, consider the intracochlear pressure measurements of Elizabeth Olsen et al. While measured at higher frequencies in the basal turn and from below the BM, the distance over which wave pressure (a component of the vertical wave) influences cochlear fluids was much larger than may be expected for the model suggested here.

12. Further to 3 and 11, how can either the reticular lamina or the tectorial membrane serve as fixed walls when both structures respond to sound in the transverse and radial directions? Please clarify this statement.

13. In addition, the authors should be more scholarly and cite papers that have seen phase changes across the organ of Corti or complexity of vibration within the organ.

Minor concerns:

1. The color coding used here makes interpretation of the figures difficult. Please consider more detailed labelling, especially in the legend of figure 3b which needs to contain information given in the legend.

2. Is it accurate to say that this series of experiments interrogates the mechanisms of low frequency hearing? The data are collected from a middle turn region closer to the basal turn than to the apical turn, in an animal whose low frequency range is rather large. This of course has no impact on the experimental results shown, only their framing and scope akin to known differences between base and apex.

Below is the list with point-by-point responses (in red) to the referees' comments. Our responses are in red.

Several of these concerns were detailed in our recently accepted JARO manuscript (Meenderink et al. 2022, JARO, available via: <https://doi.org/10.1007/s10162-022-00856-0>), which is cited in the revised manuscript. To further address these and other concerns and substantiate our conclusions, we have conducted additional experiments, doubling the number of animals for which data are included in the manuscript. These data not only fully agree with the original observations/conclusions, but they also allow us to better address (and discount) alternative explanations for the observations.

Although most of the newly obtained data are integrated in the original figures, we found it necessary to add one extra figure (Fig. 4 in revision) to the manuscript that shows: (1) the level dependence of the observed phase effect, (2) its dependence on *in vivo* conditions, and (3) its independence on exact longitudinal cochlear location. Moreover, we have updated all Figures so that they avoid the use of rainbow color maps, unifying the color scheme across the different figures. The new color scheme also makes the figures more accessible for readers with color blindness.

Reviewer #1 (Remarks to the Author):

In this paper the authors described the use of OCT for measure the direction of vibration of basilar membrane (BM) and organ of Corti (ooc). It is a relevant topic to better understand the cochlear micromechanics contribution to hearing.

Despite the relevance of the subject the aim of this paper was not well defined/stated by the author, making it difficult to the reader to identify and follow the main results. By careful reading the paper this reviewer understands that the author goal was to demonstrate that the curvature of the cochlea could be used as a method to measure the vibration direction of motion.

We have reformulated the Abstract and added to the Introduction (p. 4, l. 6–13) to clearly state the aim (i.e., determine the presence of longitudinal motion within the cochlea) of the manuscript.

It would be equivalent to change the OCT angle-of-view from previous published works. Another important aspect is related to the Methods described in the work. Due to the different steps and complexity of the analysis more details should be provided.

We have elaborated the Methods section to better describe the OCT based imaging and vibrometry (e.g., p. 8, l. 6–12), as well as the subsequent extraction of the amplitude and phase of the vibrations from these OCT signals. See also comments #5–11 by this reviewer.

From this reviewer point-of-view this work must have major revisions before being considered for publication.

Further specific points are described below.

Abstract

1 – The authors stated: “...extended region in the low-frequency apex of gerbil cochleae...”, the work described OCT images acquisition at the second-turn gerbil cochlea, not at the apex. Please clarify.

The reviewer is correct, all our recordings were from the middle turn of the cochlea. This has been corrected throughout the manuscript.

Introduction

2 – Please define BM and ooC on its the first use at the main body of the text.

Definitions of all abbreviations, including those for BM and ooC, are now provided at their first occurrence in the main text.

3 – Clearly define the goal of the work at the end of the introduction section.

We added to the Introduction section to clearly formulate the goal/aim of the manuscript (p. 4, l. 6–13).

4 – The following fragment of the last paragraph of the introduction section of the manuscript “At near-perpendicular... ...observed motions within the ooC.” describes findings from this work and should be placed at a different section (results, discussions e.g).

We have removed this part of the last paragraph from the Introduction.

Results

5 – “BM frequency response curves (FTC) were tuned (Fig. 2A) to best frequencies (BFs)” How do you define the BF? Should be clearly defined at the Methods section.

We added a description for the BF calculation to the Methods section (p. 9, l. 3–6).

6 – “We calculated wavelengths (λ) from BM phase responses across the tonotopic recording locations” How was the wavelength calculated? Should be clearly defined at the Methods section.

This was already described in the Methods section. We reformulated the description to make it clearer (p. 9, l. 7–10).

Methods:

7 – “The scan path was traversed multiple (>10) times and the intensity profiles averaged”

Does it mean that multiple B-scans were acquired at the same position? How many exactly? If changed, what was the criteria for choosing? Please rephrase and provide more details.

Yes, we acquired multiple B-scans that were then averaged. The main reason for averaging is to reduce the (background) noise floor in the images. By default, our recording software acquires sixty B-scans that are then averaged, but for some images this number was changed. Currently, it is our experience that increasing the number of B-scans in the average beyond ten does not (visually) improve the quality of the images. We have reformulated this section of the Methods section to be more precise (p. 8, l. 6–12).

8 – “Between M-scans, the angle between the...” Does it mean between adjacent M-scans?

Clearly, our original formulation was not clear. The angle between the (vertical) OCT beam and ooC varied with longitudinal location. That is, for M-scans obtained at the same longitudinal location, the angle is the same, but for M-scans from different longitudinal locations it systematically changes. We have reformulated this sentence in the Methods section.

9 – It is not clear how amplitude and phase was measured from the M-scan. Was it measured by the structural displacement? Please provide more details.

We used phase-sensitive OCT for vibrometry, which is a well-established tool that uses the phase of the OCT response at a pixel to measure its displacement over time. Unlike OCT intensity information (which is used for constructing B-scans), the OCT phase provides a resolution much better than the actual pixel size, allowing for detection of sub-nanometer vibrations (while pixel resolution is often on a micrometer scale). Extracting the magnitude and phase of the vibrations at the stimulus frequencies from the OCT phase-vs-time responses uses Fourier analysis. We have elaborated on this methodology in the Methods section (p. 8) by providing more details (e.g., the assumed refractive index) and added two references, but feel it is unnecessary to provide a detailed (mathematical) derivation of this technique.

10 – It is not clear how BF was measured. Please provide more details.

Added. See response to comment #5 by this reviewer.

11 – It is not clear how the wavelength was measured. Please provide more details.

Added. See response to comment #6 by this reviewer.

Discussions

12 – The authors presented a hydrodynamic analysis of (Fig 4) to discuss the results. It is not clear how the authors reached the conclusion using this model. One suggestion is to clearly define this analysis at the Methods and present its Results.

As experimentalists, our motivation to write this manuscript was very much driven by our experimental observations on the OHC-BM phase relation, which we found to be extremely

dependent on experimental configuration (measurement angle). In an effort to understand this observation we considered the existence of longitudinal motion (together with transverse motion), which is unavoidable in the presence of fluid surface waves. This is the reason why we introduced the hydrodynamical description of such waves; to offer a conceptual image to understand how longitudinal and transverse motions can simultaneously occur within the ooc. By no means was it intended to provide a detailed description of how the cochlea works.

We noticed that several of the referees place much emphasis on this hydrodynamical description, and how it “dictates” the interpretation in terms of exact cochlear mechanics. As said, this was not our intention for the “model”, and we considered removing it from the manuscript altogether and solely focus on the experimental findings.

However, we have decided to keep the description of the hydrodynamical analysis (which is a well-established, classical description of surface waves in fluids) because it nicely introduces the concept of longitudinal and transverse components in the motion of fluid particles. We kept it in the Discussion section of the manuscript, rather than introducing and analyzing it in the Results section since it is only used to introduce a concept that can explain the experimental observations.

Reviewer #2 (Remarks to the Author):

The authors have used the natural curvature of the cochlear partition as a means to investigate the variation of the basilar membrane and OHC-region vibration as a function of the beam axis of an OCT-vibrometry measurement system. This is a clever approach because the neither the preparation nor the OCT system need be physically moved to acquire these data. The authors provide arguments for why they are in fact measuring an effect that describes the vector nature of the velocity, and not some other confounding effect. These arguments are strong, and would be more convincing once data on the dependence of the angle of the optical axis with the BM is quantified when changing the longitudinal location (see below).

The new figure (Figure 4) shows that changing the absolute longitudinal location has no effect on the observed effect.

The authors show that a frequency-averaged-phase shift consistent with an elliptical vibration path (i.e., with both in-plane and longitudinal vector components is present – not just in-plane motion) is present– and these results are very interesting and potentially paradigm shifting.

The segregation of the positive and negative phase dependence with viewing angle is striking (shown in Figure 3A and in Figure 2E) and the authors have shown in great detail this phase result. In addition to these phase data, the amplitude of the motion as a function of the optical axis angle should be presented and analyzed with respect to consistency of the finding that there is significant longitudinal motion. This quantification may be accomplished by presenting the normalizing value of the amplitude in Fig. 2A and 2B for each angle/longitudinal location in

tabular or graphical form. This would be followed by a discussion of whether these results support your conclusions. By presenting these normalizing data, it also serves to provide the amplitude of motion for archival purposes.

The reviewer is correct that, in theory, the presence of significant longitudinal motion would also have an effect on angle-dependent amplitude ratios, not just the phase relation. However, in experiment, there are several reasons why the amplitude data are less suitable than the phase data to determine the presence of longitudinal motion.

- (1) Within a single cross-section of the organ of Corti, there is substantial variation in the response amplitudes, even when recorded from the same structure/adjacent pixels.
- (2) Variation in response amplitude is more sensitive to noise (signal-to-noise ratio) than phase variation.
- (3) The measured amplitudes do not only vary with the angle re. the longitudinal axis, but also change with the measurement angle (“ooC orientation”) in the orthogonal plane. The phase data, however, do not depend on variation in this latter angle.

Combined, these effects make that the ratio of OHC/BM amplitude varies more, independent of the “projection effect”, than the concomitant phase difference. In an initial analysis we found that the OHC/BM amplitude ratio showed more variation due to this within- and across-ooC variability than from its dependence on measurement angle. We expect that an OCT system that can truly measure motions in three dimensions is needed to verify the existence of longitudinal motion using the (relative) response amplitudes. Such a system has been under development for some time, but its application to inner-ear vibrometry has been limited thus far (e.g., Kim et al, 2022, Biomed Opt Express. 13(4): 2542–2553). Based on this, we have chosen to not try and use the amplitude data to corroborate the clear effect found in the phase data.

As for archival purposes, we have recently published a paper (Meenderink et al, J Assoc Res Otolaryngol, 2022) in which we provide a much more extensive description of the sound-evoked vibrations (amplitude and phase) across the entire organ of Corti (not limited to BM and OHC), including their dependence on stimulus frequency and intensity. This paper is now cited in the manuscript.

Change of optical axis with the BM/radial direction: The results could also change if the optical angle axis changes with respect to the BM (as shown for one optical axis by the green line and white dashed line for the radial direction in Fig. 1C). Providing an estimate for the change in this angle as a function of longitudinal location (likely a function of the helical coil angle of the cochlear ducts).

We did not measure the BM angle in the radial direction across the longitudinal axis as this angle is *not* important for the OHC-BM phase difference. If it varies with longitudinal location it will have an effect on the OHC-BM amplitude difference, but this is not reported in this manuscript.

Describe the acceptance criteria for presenting the data (e.g., discuss rationale for choosing animal #1 for 30, 50, and 70 dB while animal #3 only for 50 dB).

The acceptance criterium for including data was already stated in the Methods section of the manuscript. It is based on the phase-stability of the responses (vector strength) as assessed using Rayleigh statistics (see Figure 1 in Meenderink et al, 2022, JARO).

Since the level-dependence of the effect was not a concern when we performed these experiments (see Figures 2 & 4 in Meenderink et al, 2022, JARO, which show that phase is relatively independent of level), we simply did not acquire responses at multiple levels in all animals. The additional recordings we acquired since the initial submission provide more information on the level-dependence of the phase effect (see Figure 4).

Minor: Using FRC instead of FTC for frequency response curves makes more sense.

We replaced FTC by FRC throughout the manuscript.

Overall, this is a clearly written article with a key result that could be very important to our field. I appreciate the clear analytic presentation of the results (no over-exaggeration of the results).

Reviewer #3 (Remarks to the Author):

Review of: Organ of Corti vibrations are dominated by longitudinal motion in vivo

The novelty of this study is that longitudinal motion of the organ of Corti at the apical region of the gerbil was measured with OCT by measuring transverse-radial motion of the BM and OHCs at different longitudinal locations. As viewing angle is changed (α in Fig. 1B), the phase relationship between OHC and BM changes due to the curvature of the cochlea. The measurements of motion from OCT showed that the motions of cochlear structures are dominated by longitudinal-directed motion from the traveling wave. This is analogous to the flat elliptical motion (lots more motion longitudinally than transversely as shown in Fig. 4 C), due to the long wave with respect to depth of fluid. This study is showing that motion of the cochlear structures measured by OCT is actually dominated by this longitudinally directed motion.

This experiment is elegant, the description is clear, and the results are of interest to hearing researchers who are concerned with the mechanical motion of the structures that affect transduction of sound.

Thank you

Specific comments:

1 – Figure 1: Recommend showing naked OCT without markers and lines in addition to what is

shown. The lines and markers obstruct the view obtained by the OCT. Another picture just like C to the left of C without lines and labels. There's enough space in the lower row for this. The description in caption in (C) for the green line (A-line) is confusing, please clarify.

We have changed the figure so that the OCT intensity images are much less obscured by the added lines and markers. We felt that simply duplicating these images (e.g., once without and once with markings) would be an unnecessary use of journal space.

2 – How do you know the relative angle alpha? Also, these angle differences are small, how can you be sure of their accuracy?

The Methods section already has a description on how the angles alpha were calculated, which we slightly reformulated for clarity. It now reads (p.8 l. 31 – p.9 l. 2): “In each longitudinal OCT image, a tilted ellipse was fitted to the to the boundary of the lateral compartment of the ooc (Fig. 1B, solid white line). The angle between the normal of this ellipse and the vertical OCT beam was calculated at the longitudinal locations at which cross-sectional OCT images were obtained (Fig 1B; inset). These angles served as an estimate for the angle between the OCT beam and the BM in the longitudinal direction (termed “viewing angle α ” in the manuscript).”

With the changes made to Figure 1 (see this referee's previous remark), we tried to include a more obvious identification of the angle alpha in panel 1B. Also, the Figure's caption now includes a better description on how this angle was obtained.

We are not entirely sure what the reviewer means with the accuracy of these angles. With respect to the calculation, the accuracy is high, since they are directly calculated from the (spatial derivative) of the fitted ellipse. How well they represent the angle between the OCT beam and the longitudinal BM is not entirely know, but we estimate it to be within a few degrees. The latter inaccuracy and its effect on the interpretation is brought up in the Discussion section of the manuscript.

3 – Pg4, last para: Please be more specific in what you mean by “this”: “At multiple locations along this axis,...”. Also unclear: “...from two representative locations (8, 11, 12, 14).” What do the numbers 8, 11, 12, 14 refer to? Also what does F (and subscript numbers) mean?

“This axis” → “the tonotopic axis” See also comment #7 by this referee.

The numbers (8,11,12,14) are references. Those manuscripts also use OCT vibrometry to study intracochlear responses based on representative locations/pixels (rather than considering the responses across the entirety of the organ of Corti).

The F refers to an F-test with 1 degree of freedom and 37 observations. We reformulated this sentence to make that clearer. With the inclusion of the additional data from the three additional animals, the value of the test statistic changed, but not its significance. It now reads (p. 4, l. 27-29): In each cross-sectional image, these two locations were chosen such that the distance

between them did not systematically vary with viewing angle ($F_{1,43}=2.73$, $p=0.11$; distance mean (\pm s.d.) = 94 (± 21) μm).

4 – Number of results are minimal: #1 (responses to 3 input levels), #2 (2 input levels), #3 (only 1 input level). All 3 animals are from different distances from base. Explanation would be useful.

We obtained data in three more animals, which are now included in the revision. These additional data fully agree with the main findings and allowed us to more completely show the effect of stimulus intensity, the (in-)dependence of absolute longitudinal position, and the disappearance of the effect postmortem.

5 – Fig. 2: The legend does not have green, but the plots do. For quick interpretation, add labels “BM” to (A) and “OHC” to (B).

The colors in the figures have been changed to be consistent throughout the manuscript. With this change in colors we also avoid the use of rainbow colormaps, which are not ideal because of their inconsistent luminance. As suggested, the labels “BM” and “OHC” have been added.

6 – Fig. 2: responses are referenced to ME response. Would be good to show ME response re input sound pressure.

The presented data only have their phase referenced to the ME response [gerbil middle ear delay was well established that is 25-30 μs with a close-field sound configuration (Dong & Olson 2006)]. This is common practice when presenting cochlear vibrometry data to “remove” a small (acoustic+system) delay from the data. In our setup, with the speaker relatively close to the tympanic membrane, the ME phase-vs-frequency response curve is a (almost) straight line (=constant delay across frequencies). We believe including this as a figure in the manuscript conveys very little useful information, we now state the reason for this ME referencing in the Methods section (p. 8, l. 18).

7 – Fig. 2: Can “tonotopic axis” be better defined? Do you mean longitudinal axis?

In the cochlea, the tonotopic organization *is* along the longitudinal axis. This is why we tend to use the terms interchangeably. We realize that this may not be known by a broader audience, so we explicitly state that the tonotopic organization is along the longitudinal axis in the Introduction. Moreover, we now only use the term “tonotopic” specifically referring to frequencies, and use “longitudinal” when referring to location.

8 – Pg 5-6: Define sub-BF response. The second lower-frequency hump often seems just as big as the hump for CFs (2B). Is max magnitude always between 2-3kHz? The low-frequency hump at $\sim 1\text{kHz}$ also looks just as big for most.

For the OHC amplitude data in Figure 2, the maximum does occur between 2-3 kHz. In fact, the OHC response tuned to local BF at low sound pressure levels (see Fig. 6 in Meenderink et al,

2022, JARO). The low-frequency hump is substantial reflecting its low-pass nature, but it is smaller. We provided a better description in the Methods section how the local maximum in the amplitude curves (to determine BF) is obtained. (See also comment #5 by referee #1).

9 – Pg. 6: Refer back to Fig. 1B to remind readers of what “viewing angle” alpha means.

Done

Discussion:

10 –Please define “tonotopic axis” (same for Fig. 2 caption). “Longitudinal” might communicate better.

We replaced “tonotopic” with “longitudinal” in most places throughout the manuscript. Now, we only use “tonotopic” when explicitly referring to frequencies, otherwise “longitudinal” is used. See also comment #7 by this referee.

11 –Fig 4: Concept in A, B, C are classic motion of particles in water near the interface of air with a wave. It would be helpful to describe it that way. Also, when describing “depth”, it’s very confusing for that to be above. For A, B, C, it would be better to have it upside down. Also, it would be good to clarify in the caption that your interpretation of what is going on in the cochlea is like C.

The orientation of Figs. 5A–C, with the traveling wave at the bottom, was chosen to match our experimental condition (i.e., the BM at the bottom with the ooC on top). For this reason, we are reluctant to flip these figures upside down. For the hydrodynamical model this orientation makes no difference, the descriptions would remain the same.

12 – How does the motion observed fit with other studies, such as those that focus on the anatomy of the cochlear partition, such as the phalangeal process of linking OHCCs to dieter cells longitudinally and radially in a diagonal manner?

Yes, the highly particular anatomy within the ooC (e.g., orientation/tilt of OHC and phalangeal processes, presence of the tunnel of Corti) very likely play a role in determining the magnitude and direction of the motions. At a bare minimum, this makes the “wave medium” very anisotropic, perhaps favoring motions in the longitudinal rather than the transverse direction. Capturing the effects of ooC on the motion is far beyond the modelling effort described in this paper, it would require far more elaborate (finite-element) models of the cochlea to start capturing the anatomy and its effects on the motions. We added a sentence to the Discussion to indicate that the ooC anatomy is not captured in the simple hydrodynamical model we used, but that it can (and most likely will) alter the shape of the motion trajectories.

Reviewer #4 (Remarks to the Author):

The authors exploit the curvature of the organ of Corti to systematically vary the relative angle between portions of the OHC and BM regions, by translating their OCT beam along the tonotopic

axis of the middle turn of the cochlea. In doing so, they noted a phase grouping in the OHC region measurements, which relative to the BM varied by a lead or lag of .25 cycles and depended upon the angle of the measurement beam relative to the structures. The authors then employ a longwave model to demonstrate that such a phase transition would occur in the presence of longitudinal motion of the OHC region relative to the BM. They discuss and discount the possibility that these data may be derived from variations in the anatomy due to their use of multiple longitudinal locations as a means of varying the beam angle – which simplifies the experimental approach. They conclude by suggesting that longitudinal motion is a substantive component of organ of Corti mechanical function, and that the vector sum of OCT measurements recorded from the OHC region may be complicated by this additional, unconsidered component.

The experimental design is imaginative and the use of simple translation of the beam position may reduce the experimental error derived from manipulation of the preparation or beam angle. The ideas raised in this paper are potentially significant, thoughtful and intriguing, but more explanation and data are needed.

Major concerns and comments:

1 – Providing an n of 3 for such a novel and significant finding is insufficient. Furthermore, providing a selection of sound pressure levels from each preparation with no justification for the selection criteria makes it more difficult to understand the full picture of the data. Within animal frequency data are analyzed statistically, but between animal data are not, because there is not enough.

We have included additional data from three more animals in the revision and to reiterate, phase difference between the OHC and BM does not change with sound pressure levels (see Meenderink et al, 2022, JARO).

2 –The term OHC region is rather indefinite given that published studies are measuring vibrations of the reticular lamina (RL) and the base of OHCs at the Deiters' cell junction. These have different transverse and radial motion vectors. More specifically, from where are the measurements of this study?

Our system (and in fact in most other studies of intracochlear vibrations) does not have the spatial resolution to claim that recordings were made from the top of reticular lamina, or precisely at the base of the OHC (OHCs at the Deiters' cell junction). Across our recordings, the "OHC recordings" were on average 94 micrometer away from the BM. This places this recording location within the OHC region, but most likely not exactly at their base or apex (reticular lamina). We therefore use the term "OHC region" within the manuscript. In a recently accepted manuscript (Meenderink et al, JARO 2022), we present vibrometry data across the OHC region which shows that with the phase of these responses is constant along OHC basal-to-apical axis. With this, the exact location within the OHC region is not important; the phase is the same throughout.

3 –The hypothesized elliptical motion is set against the fixed boundary of the RL and tectorial

membrane (see 11 below for more about the boundary). Does this imply Rayleigh type water waves? In that case wouldn't the ellipse be oriented with the long axis in the transverse direction the greatest movement be near the basilar membrane?

Rayleigh surface waves are different from water waves in that the latter occur in solids. These are not described in the manuscript. For water waves the elliptical motion within the wave medium is oriented in a plane parallel to the propagation direction of the TW. As shown in Figure 5, the relative amplitudes of the longitudinal (in the direction of the TW) and the transverse (orthogonal to the TW) motion depends on the wavelength, depth of the fluid and the distance away from the TW-supporting layer, but the longitudinal motion is never smaller than the transverse motion.

4 – Further to 3, would the nature and location of the elliptical motion be a way to test the hypothesis? Firstly, there would be a graded effect with depth. Secondly some Rayleigh waves have reversal of the elliptical direction with depth. See also point 11.

Even if the cochlea's hydrodynamics were fully described by the simple model we used and recording at different (transverse) distances away from the BM were obtained, we believe it would not be trivial to determine the (changes in the) shape of the elliptical motion since only the projection of this motion onto the measurement axis is captured. The anisotropy of the oOC, the BM orientation, and spatial resolution of the system would make such experiments exceedingly difficult...

5 –Figure 1 shows the direction of light beam view to be passing through the Hensen's cells when observing the OHC region. Hensen's cells are often the exact location of vibration measurements in studies because of the high degree of reflectance. Even when the OCT coherence gate is located at the OHC region the measured vibration can be a combination of the gate and of out of gate vibration. Hensen's cells could even dominate. Might Hensen's cells have a longitudinal vector component and view direction result in a phase flip? Resolving such questions requires more data on the transverse and radial motion of the OHC region.

In our experiments, the OHC region tends to have higher reflectivity than the cells in the overlying lateral compartment (e.g., Hensen's cells, see Fig. 1C), and it is to be expected that the OHC vibrations would dominate over the latter responses. We have documented lateral compartment vibrations in detail (Meenderink et al, JARO 2022), and found them to be distinct from both OHC and BM vibrations. Finally, we have some preliminary data showing that the phase difference between the BM and the Hensen's cells does not vary with viewing angle. Combined, these observations strongly suggest that the OHC responses are not "contaminated" by responses from other pixels along the recorded A-line.

6 –. The OHC region is said to be measured here and this phase transition occurs at the OHC place, one may surmise that longitudinal motion may be enhanced by amplification and as such the phase transition is a physiologically vulnerable process. No such data (such as post mortem measurements) was provided to confirm or refute this.

We now present postmortem data (Fig. 4) and show that the 0.5-cycle phase transition observed *in vivo* is not present in the passive cochlea. The whole cochlear partition moved almost in-phase after the animal died (also see Fig. 2C and E in Meenderink et al, 2022, JARO)

7 – Further to 6, the phase transition shown for animal #1 in figure 3 indicates that the magnitude of the phase transition might show some level dependence. Since there is just one 30 dB SPL record shown, it is difficult to know if this is repeatable.

The additional recordings we now include in the manuscript were obtained over the full range of stimulus intensities (30 to 70 dB SPL). The phase difference between OHC and BM showed no evidence for a level dependence in the observed effect (e.g. Fig. 4).

8 – The authors declare that the phase flip is entirely dependent upon the sign of the viewing angle. This is not borne out by the data – animal #1 in figure 3a flips for a still negative viewing angle. It is difficult to tell if animal #2 flips with sign of the angle as there appears to be missing data around the 0 point for this animal. Animal #3 does follow the expected behavior. This variability highlights a requirement for more data presented more consistently across angle, level and frequency.

The referee is absolutely correct that the data may suggest (see Fig. 3A) that the phase flip occurs for slightly negative viewing angles (between 0 and -5°). We do believe that this would not discount the interpretation. For one, determining the viewing angle is likely subject to some error, as we describe in the Discussion. Moreover, a slight “tilt” (re. the longitudinal TW direction) in the elliptical trajectories would result in the phase flip to occur at a negative angle. Such a tilt may arise when the wave medium is anisotropic. This is described in more detail in Cooper et al., 2018 (cited in the manuscript).

We elected to not introduce this elliptical tilt in the manuscript to not to over-complicate the description and interpretation of the introduced hydrodynamical description. The model is not intended to explain the data in great detail, it simply serves to introduce the concept of elliptical fluid motion.

9 – In figure 2 E, the relative phase between the two structures is shown as a function of frequency. For each curve shown there is a phase variation as frequency increases, and the direction of this variation depends upon the sign of the viewing angle. For negative angle measurements, the high frequency phase difference trends towards 0, so the two structures now move in phase. However for positive angle measurements the same phase difference is now magnified. If there is a longitudinal component, there is clearly some frequency based variation of its behavior. Is it possible to reconcile the angle sign dependent widening phase difference vs converging phase difference observed here?

The frequency dependence of the phase difference as seen in Figure 2 was not consistently observed across animals. In several animals, no obvious frequency dependence was observed at all, while others showed an opposite trend, where the phase difference tended to zero for the

lowest stimulus frequencies. We felt it to be outside the concept of the manuscript to fully characterize the frequency dependence. This would require many more experiments in which frequency, location, and viewing angle are varied independently.

Having said this, some frequency dependence of the phase effect (and OHC-BM phase difference) may be expected. The shape of the ellipses depends on wavelength, which systematically decreases with frequency (e.g. Fig. 2G) such that for higher frequencies, responses may be more like short waves (Fig. 5A), rather than the long waves (Fig. 5C).

10 – Also, looking at Fig. 2F, one can see that the BF for the BM and OHCs differs in a rather dramatic fashion. It seems to be lower for the OHCs. The problem here is that the legend says the data were acquired at 30 dB SPL, but only one animal had data at this sound pressure level. So despite the error bars, it seems to be an n of 1 here? And then the linear fit is calculated for all the points. In short, this is rather confusing and adds to the need for more experiments.

Yes, Fig 2A-F show data for one animal. In this figure, only panel G shows data across 6 animals. As stated in the manuscript, the error bars in Fig. 2F represent the uncertainty in deriving the BF in this one animal. For the presentation and interpretation of the observed phase effects, the exact longitudinal locations (and thus BF's) turn out not to be important (e.g. Fig. 4), and are not presented for the other five animals (although they are very similar since we recorded from approximately the same cochlear location in all animals).

11 – The authors state that elliptical motion where the major component is longitudinal produces a shallow wave profile, limiting the extent of the wave's pressure influence in the transverse direction to the boundaries of the organ of Corti. Is the effect subject to a form of scaling symmetry? Would the shorter wavelength of a higher frequency wave still be dominant to the transverse displacement? If so, the transverse wave depth would be even smaller. How can this be explained in the context of the observed motion of Reissner's membrane, which is assumed to follow the motion of the organ of Corti? Additionally, consider the intracochlear pressure measurements of Elizabeth Olsen et al. While measured at higher frequencies in the basal turn and from below the BM, the distance over which wave pressure (a component of the vertical wave) influences cochlear fluids was much larger than may be expected for the model suggested here.

The traveling wave is supported by the cochlear partition and fluid inertia, therefore, it can be measured within the cochlear fluid in the format of intracochlear pressure and also from the cochlear partition as displacements/velocities, i.e., the reticular laminar, the BM and OHC regions. Over the years, our explanation has been focused on the transverse up-and-down motion of the cochlear partition, i.e., the BM, but paid a little attention to the real three-dimensional motion of the system including the longitudinal motion that in-and-out of the plane of a cross-section of the cochlea.

The long- or short-wavelength is defined relative to the height of the scala in 2D or 3D models. From our measurements, the wavelength at the BF (~ 1mm) was greater than the height of the

scala (~ 0.5 mm from BM to Reissner's membrane, see Fig. 1C), thus, at least in this frequency region, the wave propagation appears to be in the long wave region. This may explain the difference between the low- and high-frequency region (at basal turn, the wavelength was about 1/3 mm, shorter than the height of the scala. See de La Rochefoucauld O, Olson ES. *Biophys J.* 2007).

12 – Further to 3 and 11, how can either the reticular lamina or the tectorial membrane serve as fixed walls when both structures respond to sound in the transverse and radial directions? Please clarify this statement.

We like to reiterate that the experimental observations are consistent with elliptical motion trajectories (e.g. the presence of a longitudinal component). If the cochlear responses were described well by a surface wave in a (isotropic) fluid, the results suggest that the longitudinal motion is much larger than the transverse one. The presence of the ooC undoubtedly severely affects the accuracy of this interpretation and several mechanisms can be imagined by which the BM motions can “escape” the ooC. For example, we have preliminary data that the more radially located portion of the ooC (Hensen/Claudius) cells seem to exhibit much smaller longitudinal motion and may thus serve as an “escape route” for BM motions to reach scala media. Alternatively, the TM or reticular lamina may not be a “fixed wall”, but another interface along which a TW may occur from which elliptical motions penetrate into scala media.

These explanations are rife with speculation, and we elected to use the hydrodynamical description in Fig. 4 to only introduce the concept of elliptical motion...

13 – In addition, the authors should be more scholarly and cite papers that have seen phase changes across the organ of Corti or complexity of vibration within the organ.

There are only a few papers that systematically measured sound-evoked vibrations within/across the ooC. These manuscripts are cited in the Introduction in this context (p. 3, l. 25).

Minor concerns:

1 – The color coding used here makes interpretation of the figures difficult. Please consider more detailed labelling, especially in the legend of figure 3b which needs to contain information given in the legend.

The figures and their color coding have been updated. The latter is now identical across the different figures.

2 – Is it accurate to say that this series of experiments interrogates the mechanisms of low frequency hearing? The data are collected from a middle turn region closer to the basal turn than to the apical turn, in an animal whose low frequency range is rather large. This of course has no

impact on the experimental results shown, only their framing and scope akin to known differences between base and apex.

We now no longer claim to study low frequency hearing. Rather, we simply state that recordings were from the middle turn in the gerbil cochlea.

Reviewers' comments:

Reviewer #1 (Remarks to the Author):

I used a color coded scheme to facilitate the reviewing process.
The comments for the authors are in the document attached.

Reviewer #2 (Remarks to the Author):

The important result of this paper is that the relative phase of the BM and OHC region depends on the angle of the measurement beam, and the interpretation of the results that arise from OCT measurements of the ooC must be handled very carefully. They have identified a phase "flip" from a quarter cycle lead to a quarter cycle lag – critical for the interpretation of energy generation. Moreover, they show that small angular changes give rise to this change. The experimental approach is clever and impressively repeatable across animals. These experimental results and their processing will be of high impact on the field. Two problematic elements of the paper remain

First, the authors make a few claims and justifications of the results regarding a surface wave. This argument is an example, but the analogy is taken too far in my opinion. There is no justification or verification that surface waves of this sort are present in the ooC. In the complex ooC microstructures it is completely justifiable to say that the polarization of the BM and OHC motion locations are different (by polarization, I mean the ratio of the vector components in the three directions, transverse, radial, and longitudinal). Again, in my opinion the surface wave analogy is good to mention as part of the justification (but just part – as the elements/components/structures may simply have different polarizations). I find Figs. 5A, B, and C to be distracting and confusing and should (along with most of the discussion of surface waves) be removed. Fig. 5D shows the key, direct, and important message of the paper.

In the Abstract, the authors mention the "unavoidable presence of surface waves". However, in the paper no evidence of surface waves in/on the ooC is given – but evidence for longitudinal velocity component structures is given. Lines 167-183 are problematic. Reviewer 4 commented on the difficulty in labelling the TM or RL as the "fixed wall" for the surface wave, and I agree – I strongly advocate for removing this element (this whole paragraph) from the paper. As the authors mention, the surface wave analogy is a simplification (unnecessary from my view), that will like confuse future researchers if taken as far as currently in the paper.

Second, important documentation is needed: In Fig. 2 A and 2B, the amplitudes are normalized by the maximum vibration. These normalization values should be given in the caption of the figure for each curve. This provides the reader and future replicator of the data a point of reference for the results.

Minor question: I think the following is a small effect, but I would like the authors to comment on the effect (if any) of the pitch angle of the cochlear spiral (this is different from the radius of curvature – this is the angle of inclination or height gain – as a function of the longitudinal coordinate around the spiral) has on your analysis – this could also cause a change in phase (and one that would be seen if the analysis window was rotated by 10 degrees – as was done in Figure 4).

Minor clarification:

Line 102 – representative locations 8,11,12, 15 – what do these numbers mean?

Reviewer #5 (Remarks to the Author):

The authors have addressed almost all of the reviewer comments in a scholarly and thorough way and the manuscript is much clearer. Only one question remains, brought about by their kind inclusion of useful new data at reviewer request: the post-mortem data shows that there is no phase flip with viewing angle, i.e. no longitudinal motion in the passive preparation. However, the authors also report no evidence of level dependence of this phenomenon i.e. the saturation of the OHC force component of this motion does not change the behavior of the longitudinal motion. Regarding basic tenets of the non-linear cochlea vs the passive, these two concepts do not agree. It would be expected that high level stimulation should eliminate or minimize phase transitions in a similar manner to the loss of amplification caused by death. The reviewer does not get the impression that this has been discussed in the manuscript. Instead, the authors discuss the possibility of the absence of saturation for OHC force production, both here and in Meenderink et al. 2022. In the latter paper, some discussion of level dependence is made but not with respect to its physiological vulnerability, only with respect to other observations in the literature. Furthermore, the concept is modelled in figure 3, where a unity amplitude ratio between the two measurement locations reveals no phase transition - akin to a dead cochlea or one possibly one undergoing high levels of stimulation. How then do the authors reconcile this possible disagreement?

Below is the list with point-by-point responses to the referees' comments. Our responses are in red.

Reviewer #1 (Remarks to the Author):

I used a color coded scheme to facilitate the reviewing process. The comments for the authors are in the document attached.

In the attached document, the reviewer raised three questions that are repeated and answered below.

Minor issue:

- (1) Line 208 – 210: “Acquisition of optical spectra was controlled by externally generated TTL pulses (rate: 27.9 pulses/ms) that were synchronized to the stimulus generation and microphone acquisition system (RX6: Tucker Davies Technologies system III).”

There were any delays between the stimulus and OCT image acquisition? How this synchronization affect the OCT measurement?

Yes, there is a (small) delay between the stimulus (as it exists inside the computer) and the OCT image acquisition, but it does not affect the results.

The delay comes from a difference in the “speed” of D/A conversion and Digital Out generation of the hardware. This delay is constant and is easily adjusted for so that it does not affect the OCT measurements. More importantly, results presented in the manuscript are relative measurements in which the difference between two recordings (e.g., BM vs OHC) is used. In this case, any constant delay shared between these recordings is cancelled.

- (2) Lines 212 – 210: “Intensity images (B-scans) were constructed from 1024 A-lines obtained at fixed distances along a “scan line”. To improve the quality of these images we averaged at least 60 B-scans, where each B-scan was completed prior to acquisition of the next one.”

What was the field-of-view of each B-scan?

We are not entirely sure what the reviewer means by “field-of-view of each B-scan” but assume he/she refers to the width of the B-scan images. This differed across animals/images but was between 1.6 and 3 mm for longitudinal images (e.g., Figure 1B) and 0.6 and 1 mm for orthogonal images (e.g., Fig. 1C). We included this information in the manuscript by adding the following sentence to the Materials section (lines 200-201):

The length of the scan line was 1.6–3 mm for parallel images and 0.6–1 mm for orthogonal images.

- (3) Lines 218 – 219: “...Here, the OCT measurement beam was kept at a fixed position, and a series of time-stamped A-lines was recorded...”

How many A-scan were used in the M-scan and what is the exposure time? How it is related to the first question (synchronization).

The number of A-scan in each M-scan depends on the stimulus duration. As described in *Materials & Methods: Acoustic stimulation and analysis*, the stimulus duration varied between 36 and 60 seconds (depending on stimulus intensity), which corresponds to 1,004,463 and 1,674,105 A-lines, respectively. The exposure time for each A-line was ~28 microsec. This is unrelated to the synchronization in the first question.

Reviewer #2 (Remarks to the Author):

The important result of this paper is that the relative phase of the BM and OHC region depends on the angle of the measurement beam, and the interpretation of the results that arise from OCT measurements of the ooC must be handled very carefully. They have identified a phase “flip” from a quarter cycle lead to a quarter cycle lag – critical for the interpretation of energy generation. Moreover, they show that small angular changes give rise to this change. The experimental approach is clever and impressively repeatable across animals. These experimental results and their processing will be of high impact on the field. Two problematic elements of the paper remain

First, the authors make a few claims and justifications of the results regarding a surface wave. This argument is an example, but the analogy is taken too far in my opinion. There is no justification or verification that surface waves of this sort are present in the ooC.

In the complex ooC microstructures it is completely justifiable to say that the polarization of the BM and OHC motion locations are different (by polarization, I mean the ratio of the vector components in the three directions, transverse, radial, and longitudinal). Again, in my opinion the surface wave analogy is good to mention as part of the justification (but just part – as the elements/components/structures may simply have different polarizations). I find Figs. 5A, B, and C to be distracting and confusing and should (along with most of the discussion of surface waves) be removed. Fig. 5D shows the key, direct, and important message of the paper.

In the Abstract, the authors mention the “unavoidable presence of surface waves”. However, in the paper no evidence of surface waves in/on the ooC is given – but evidence for longitudinal velocity component structures is given. Lines 167-183 are problematic. Reviewer 4 commented on the difficulty in labelling the TM or RL as the “fixed wall” for the surface wave, and I agree – I strongly advocate for removing this element (this whole paragraph) from the paper. As the authors mention, the surface wave analogy is a simplification (unnecessary from my view), that will like confuse future researchers if taken as far as currently in the paper.

Following the reviewer’s suggestion we have substantially reduced the role of the hydrodynamical model in the manuscript. We now only refer to it via citation of previous cochlear models, and simply state that (lines 148-151):

Descriptions of the hydrodynamics of the cochlea^{13,14,27} predict that particle motion some distance away from the BM is along ellipses, with displacements in the longitudinal direction that lag the transverse vibrations by 0.25 cycle.

With this reduction we have also changed Fig. 5 in the manuscript by removing panels A-C (as suggested by the reviewer), which gave a graphical illustration of the hydrodynamical model.

We also acknowledge the reviewer's observation that the complex ooC microstructure may also contribute to the occurrence of longitudinal motion and added (lines 161-165):

"Anisotropies in the ooC are likely to complicate the relatively simple hydrodynamical description of the cochlea and its predictions of (the orientation and shape of) the elliptical fluid motion. For example, the geometry of the OHC/Deiter cell and its phalangeal process/tunnel of Corti may favor longitudinal re. transverse vibrations, resulting in flatter ellipses than expected from the ratio between wavelength and wave medium depth alone."

Finally, we have completely removed the section about the potential role of the TM or RL as the "fixed wall".

With these changes, we felt it necessary to reformulate parts of the abstract to remove the statement "...unavoidable presence of surface waves". Rather, we now only state that: "We interpret these results as evidence for significant longitudinal motion within the ooC..." without any mention of surface waves or the hydrodynamical model.

Second, important documentation is needed: In Fig. 2 A and 2B, the amplitudes are normalized by the maximum vibration. These normalization values should be given in the caption of the figure for each curve. This provides the reader and future replicator of the data a point of reference for the results.

We have added the absolute values at BF for all curves into Fig 2A and 2B. This is now described in the Figure's caption.

Minor question: I think the following is a small effect, but I would like the authors to comment on the effect (if any) of the pitch angle of the cochlear spiral (this is different from the radius of curvature – this is the angle of inclination or height gain – as a function of the longitudinal coordinate around the spiral) has on your analysis – this could also cause a change in phase (and one that would be seen if the analysis window was rotated by 10 degrees – as was done in Figure 4).

The longitudinal images (e.g., Fig. 1B) were taken such that they "follow" the pitch of the cochlear spiral. As a consequence, this pitch will only have a (presumably) small effect on the BM angle in the radial direction. This angle, however, is not important for the observed phase effect.

Minor clarification:

Line 102 – representative locations 8,11,12, 15 – what do these numbers mean?

These numbers are references to papers.

Reviewer #5 (Remarks to the Author):

The authors have addressed almost all of the reviewer comments in a scholarly and thorough way and the manuscript is much clearer. Only one question remains, brought about by their kind

inclusion of useful new data at reviewer request: the post-mortem data shows that there is no phase flip with viewing angle, i.e. no longitudinal motion in the passive preparation. However, the authors also report no evidence of level dependence of this phenomenon i.e. the saturation of the OHC force component of this motion does not change the behavior of the longitudinal motion. Regarding basic tenets of the non-linear cochlea vs the passive, these two concepts do not agree. It would be expected that high level stimulation should eliminate or minimize phase transitions in a similar manner to the loss of amplification caused by death. The reviewer does not get the impression that this has been discussed in the manuscript. Instead, the authors discuss the possibility of the absence of saturation for OHC force production, both here and in Meenderink et al. 2022. In the latter paper, some discussion of level dependence is made but not with respect to its physiological vulnerability, only with respect to other observations in the literature. Furthermore, the concept is modelled in figure 3, where a unity amplitude ratio between the two measurement locations reveals no phase transition - akin to a dead cochlea or one possibly one undergoing high levels of stimulation. How then do the authors reconcile this possible disagreement?

We agree with the reviewer that (high-level) responses in the active (live) and (passive) cochlea are not the same. This, however, is not a “disagreement”. There is ample evidence in literature that responses in the dead cochlea differ from high-level responses in the living cochlea. For example, the figure below is from Figure 3 from Dewey et al. (2018). Cell Rep 5: 2915-2927 and directly compares frequency response curves (FRC) in live and dead animals. Both the gain and the phase of these FRC differ in several respects. For example, responses in the live cochlea grow compressive to the highest stimulus intensities (80 dB SPL), while compression is absent in the dead cochlea (panels A-D). Even at this high intensity, the dead and alive FRC are not the same (panels I-L). Moreover, exploring the (absence of) level-effects on the observed phase effect was not a focus in the current manuscript, and we refrained from including it in the Discussion.

Figure 3 from Dewey et al. (2018). Cell Rep 5: 2915-2927

REVIEWERS' COMMENTS:

Reviewer #2 (Remarks to the Author):

The authors have addressed my comments in a thoughtful and thorough way. This will be an excellent contribution to the literature. I recommend the paper be published.

Reviewer #5 (Remarks to the Author):

I have no further comments and think this paper is sufficiently revised for publication.